

**Projected future changes in cryosphere and hydrology of a**
**mountainous catchment in the Upper Heihe River, China**
Zehua Chang [1], Hongkai Gao [1]*, Leilei Yong [1], Kang Wang [1], Rensheng Chen [2],
Chuntan Han [2], Otgonbayar Demberel [3], Batsuren Dorjsuren [4], Shugui Hou [5], Zheng
Duan [6]
[1] Key Laboratory of Geographic Information Science (Ministry of Education of
China), School of Geographical Sciences, East China Normal University, Shanghai,
China
[2] Qilian Alpine Ecology and Hydrology Research Station, Key Laboratory of
Ecohydrology of Inland River Basin, Northwest Institute of Eco-Environment and
Resources, Chinese Academy of Sciences, Lanzhou 730000, China
[3] Department of Geography and Geology Khovd branch of National University of
Mongolia, Erkh choloonii street, Khovd, Mongolia
[4] Department of Environment and Forest Engineering, National University of
Mongolia, Ulaanbaatar 210646, Mongolia
[5] School of Oceanography (SOO), Shanghai Jiao Tong University (SJTU), Shanghai,
China
[6] Department of Physical Geography and Ecosystem Science, Lund University,
Sölvegatan 12, SE-223 62, Lund, Sweden
*Correspondence: Hongkai Gao (hkgao@geo.ecnu.edu.cn)
**Abstract**:Climate warming exacerbates the degradation of the mountain cryosphere,
including glacier retreat, reduction in snow cover area, and permafrost degradation.
These changes dramatically alter the local and downstream hydrological regime,
posing significant threats to basin-scale water resource management and sustainable
development. However, there is still a lack of systematic research that evaluates the
variation of cryospheric elements in mountainous catchments and their impacts on



future hydrology and water resources. In this study, we developed an integrated
cryospheric-hydrologic model, referred to as the FLEX-Cryo model. This model
comprehensively considers glaciers, snow cover, frozen soil, and their dynamic
impacts on hydrological processes in the mountainous Hulu catchment located in the
Upper Heihe river of China. We utilized the state-of-the-art climate change projection
data from the sixth phase of the Coupled Model Intercomparison Project (CMIP6) to
simulate the future changes in the mountainous cryosphere and their impacts on
hydrology. Our findings showed that the two glaciers in the Hulu catchment will
completely melt out around the years 2045-2051. By the end of the 21st century, the
annual maximum snow water equivalent is projected to decrease by 41.4% and
46.0%, while the duration of snow cover will be reduced by approximately 45 and 70
days. The freeze onset of seasonal frozen soil is expected to be delayed by 10 and 22
days, while the thaw onset of permafrost is likely to advance by 19 and 32 days.
Moreover, the maximum freeze depth of seasonal frozen soil is projected to decrease
by 5.2 and 10.9 cm per decade, and the depth of the active layer will increase by 8.2
and 15.5 cm per decade. Regarding hydrology, runoff exhibits a decreasing trend until
the complete melt-out of glaciers, resulting in a total runoff decrease of 15.6% and
18.1%. Subsequently, total runoff shows an increasing trend, primarily due to an
increase in precipitation. Permafrost degradation causes the duration of low runoff in
the early thawing season to decrease, and the discontinuous baseflow recession
gradually transitions into linear recessions, leading to an increase in baseflow. Our
results highlight the significant changes expected in the mountainous cryosphere and
hydrology in the future. These findings enhance our understanding of cold-region
hydrological processes and have the potential to assist local and downstream water



resource management in addressing the challenges posed by climate change.
**Keywords**: Glacier, Snow cover, Seasonal frozen soil, Permafrost, Runoff, Model
prediction

**1.  Introduction**

"How will cold region runoff and groundwater change in a warmer climate?"

was identified by the International Association of Hydrological Sciences (IAHS) as
one of the 23 unsolved scientific problems (Blöschl et al., 2019). The mountain
cryosphere, which includes glaciers, snow cover, and frozen soil in high-altitude
regions, has a significant impact on water resources (Adler et al., 2019; Arendt et al.,
2020; Rasul et al., 2020; Zhang et al., 2022). Mountain cryosphere is considered a
crucial "water tower" and a climate change indicator due to its sensitivity to climate
change (Tang et al., 2023). However, the cryosphere is rapidly retreating in many
parts of the world, including glacier retreat, expansion of glacier lakes, northward
movement of the permafrost southern limit, and shrinking snow cover area (Moreno
et al., 2022; S. Wang et al., 2022; Ding et al., 2019; Wang et al., 2023). These changes
have disrupted the water tower region and pose significant challenges to sustainable
water resources management (Ragettli et al., 2016; Yao et al., 2022).

The degradation of the mountain cryosphere varies from region to region

(Andrianaki et al., 2019; Wang et al., 2019). Lower altitudes experience a decreasing
trend in snow cover days, snow depth, snow water equivalent, and snowmelt due to
climate warming, while higher altitudes present a more complex picture (Connon et
al., 2021; Nury et al., 2022; Yang et al., 2022). Global continental glacier mass
balance from 2006 to 2015 was approximately -123±24 GT yr$^{-1}$, with significant





losses observed in the Southern Andes, Caucasus Mountains, and Central Europe,
while the Karakoram and Pamir regions exhibited lesser loss (Intergovernmental
Panel on Climate Change (IPCC), 2022; Van Der Geest and Van Den Berg, 2021).
Future projections suggest a 40% decrease in global permafrost by the end of the
century, potentially transitioning into seasonal frozen soil (Chadburn et al., 2017). The
mountain cryosphere serves as a significant freshwater reservoir, impacting water
resources and the hydrological cycle (Ding et al., 2020).

In a warming climate, glacier runoff exhibits a "tipping point" characterized by

an initial increase followed by a subsequent decline (Rosier et al., 2021; Zhang et al.,
2012). While small glaciers have already experienced this tipping point, its
occurrence in large glaciers remains uncertain (Brovkin et al., 2021; Huss and Hock,
2018). Permafrost degradation leads to an increase in active layer thickness, resulting
in the melting of subsurface ice and an augmentation of soil water storage capacity
(Abdelhamed et al., 2022). Additionally, the degradation of the cryosphere
significantly impacts the atmosphere, biosphere, surface energy balance, ecological
water use, and ecosystems (Gilg et al., 2012; Miner et al., 2022; Pothula and Adams,
2022). Understanding the complex interactions between cryosphere degradation and
ecosystems is crucial, but quantitatively observing the degradation process in high-
altitude regions is challenging. Hydrological models provide an effective approach to
analyze degradation patterns and assess the impact on future water resources (Han and
Menzel, 2022).

Glacio-hydrology is influenced by both glacier melt and glacier dynamics.

Glacier melting models can be categorized into three types: energy balance,
temperature index, and hybrid models (He et al., 2021; Gao et al., 2021; Negi et al.,



2022; Zekollari et al., 2022). While energy balance models analyze glacier
accumulation and melt processes based on solid physical mechanisms, they require
extensive forcing data that may not be readily available in mountainous regions (Huss
et al., 2010). On the other hand, temperature index models are simpler and more
effective, requiring fewer parameters (including degree-day factor and threshold
temperature) and forcing data (temperature and precipitation) (Bolibar et al., 2022;
Vincent and Thibert, 2023). It performs well at both daily and monthly scales.
Glaciers are moving slowly, due to the combined effects of gravity and high viscosity
of ice. Due to climate change, ice becomes thinner, and glacier loses its mass balance,
which will cause the glacier morphology evolve to a new balance status. Glacier
dynamic models, with full-Stokes approach as the most complete form, and many
other simplifications, such as the shallow-ice approximation, and the shallow-shelf
approximation etc, are still computationally expensive, hindering their implications in
large scale studies. Three conceptual models are commonly used for glacier evolution:
volume-area scaling (V-A) method, accumulation area ratio (AAR) method, and $\Delta$h-
parameterization (Michel et al., 2022; Wiersma et al., 2022). The first two approaches
do not consider the detailed changes in different elevation bands, while the $\Delta$h-
parameterization approaches only require glacier mass balance as forcing data to
analyze changes in ice thickness at different elevation bands based on the relationship
between glacier mass balance and glacier area (Huss et al., 2010). The temperature
index method coupled with the $\Delta$h-parameterization approach serve as effective
module to simulate glacier evolution and its impacts on hydrology.

Permafrost hydrology models can be classified into one-dimensional models and

distributed watershed models (Elshamy et al., 2020). One-dimensional hydrological



models, such as the Stefan equation, the temperature at the top of permafrost (TTOP)
model, CoupModel, and SHAW model, are effective in simulating freeze depth,
hydrothermal transport, and carbon or nitrogen transport, but they are unable to
capture the broader impact of permafrost on hydrology at catchment scale (Kaplan
Pastíriková et al., 2023; Li et al., 2022; Liu et al., 2023). On the other hand,
distributed watershed models, such as the Cold Regions Hydrological Model
(CRHM), Hydrogeosphere (HGS), and Distributed water-heat coupled model
(DWHC), consider the spatial variability of permafrost properties and simulate the
interactions between permafrost, surface water, and groundwater (Chen et al., 2008;
He et al., 2023; Pomeroy et al., 2022). These models operate on a small-scale basis
and require extensive prior knowledge, following a "bottom-up" approach that relies
on small-scale field observations and situational models to comprehend the effects of
permafrost on hydrology. However, the freeze-thaw cycle is influenced by multiple
interconnected factors, including climate, topography, slope orientation, snowpack,
and vegetation (Chang et al., 2022). The process of upscaling would lead to the
neglect of some variables and the amplification of others (Fenicia and McDonnell,
2022). In contrast, the FLEX-Cryo model is based on the FLEX-Topo-FS model,
which employs a "top-down" modeling procedure that involves observed data
analysis, qualitative perceptual modeling, quantitative conceptual modeling, and the
testing of model realism. This model exhibits the ability to accurately and
expeditiously identify key elements in permafrost hydrological processes and then
simulate hydrology at the catchment scale (Gao et al., 2022).

The aim of this study is to integrate the FLEX-Topo-FS model and a glacier

evolution model (Δh-parameterization) to develop a landscape-based model of the



mountain cryosphere, referred to as FLEX-Cryo. This model will be utilized to
simulate changes in various components of the mountain cryosphere and evaluate
their impacts on hydrological processes, thereby enhancing our understanding of the
hydrological cycle. The model will be driven by eight bias-corrected Global Climate
Models (GCMs) under SSP2-4.5 and SSP5-8.5 scenarios obtained from the Coupled
Model Intercomparison Project Phase 6 (CMIP6), which will be used to predict future
changes in glaciers, snow, and frozen soil, as well as their effects on hydrology.
**2.Study area and data**
**2.1 Study area**

The Hulu catchment is located in the upper reaches of Heihe River basin (38°

12′ N-38° 17′ N, 99° 50′ E-99° 53′ E) and about 23.1 km². The elevation
ranges from 2960-4820m. The Hulu catchment belongs to continental monsoon
climate. Rainfall is the major phase of precipitation, and there is also snowfall in the
winter. Four landscapes are identified, i.e. glacier (5.6%), alpine desert (53.5%),
vegetation hillslope (37.5%), and riparian zone (3.4%; Fig.2). The landscape pattern
in Hulu catchment has typical altitude zonality. The vegetation and riparian are almost
distributed in the lower elevation bands. Alpine desert, and glacier are in the high
elevation bands. There is almost no human activity in the catchment. There are two
glaciers, i.e. Glacier1 and Glacier2 (Fig.1) in the catchment. And the Galcier1 was
also named as the Shiyi Glacier in the glacier catalogue of China. Seasonal frozen soil
and permafrost both exist in the catchment. The lower limit of permafrost is around in
3650-3700 m. Permafrost region account for 64% of the total catchment and the
others are seasonal frozen soil. The soil generally starts to freeze in the October. Thus
October 1 was set as the start of hydrological year, so forth. All the interannual



variations in this study were based on the hydrology year.

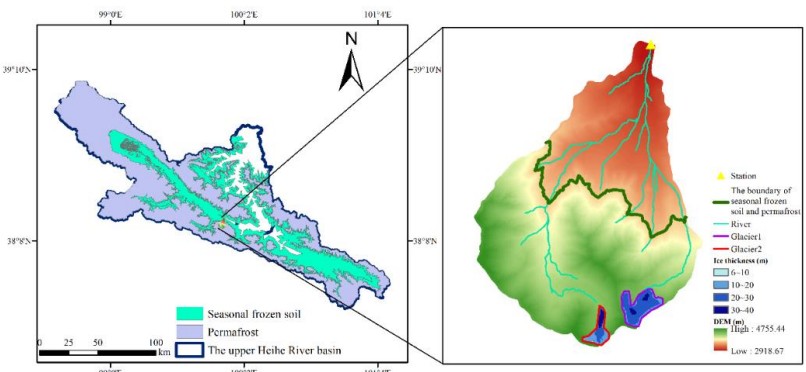


Figure 1. The digital elevation model, the thickness of glacier and the location of
study area

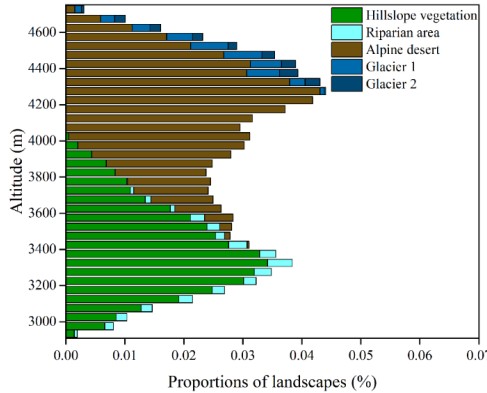


Figure 2. Landscape classification at different elevation bands
**2.2 Data**
Temperature and precipitation are observed at 2920 m, near the outlet of the
catchment, from January 1, 2011 to December 31, 2014. Farinotti et al. (2019) used
five models which used the ice flow dynamics to invert ice thickness from surface
features to estimate the ice thickness distribution of about 21500 glaciers outside the
Greenland and Antarctic ice sheets. We used the estimated data for the initial
thickness distribution of Glacier1 and Glacier2 (data download from



https://doi.org/10.3929/ethz-b-000315707).
The Couple Model Intercomparison Project phase 6 (CMIP6) is widely used to
predict future climate. Eight general circulation models (GCMs) (Table 1) under two
climate scenarios (SSP2-4.5 and SSP5-8.5) are used for predicting future climate.
SSP2-4.5 scenario represents medium part of the future pathways, which is usually a
referenced experiment comparing others CMIP6-Endorsed MIPs and it produces a
radiative forcing of 4.5 W m$^{-2}$ in 2100. SSP5-8.5 scenario represents the high
emission scenario and it produce a radiative forcing of 8.5 W m$^{-2}$ in 2100.
There is certain bias in the output of GCMs that needs to be corrected. Firstly,
outputs from eight GCMs under two climate scenarios are interpolated to 0.5°×0.5°,
then the bias corrects are carried out by CMhyd software (download from
https://swat.tamu.edu/software/cmhyd/) in which four methods were used including:
distribution mapping of precipitation and temperature, linear scaling of precipitation
and temperature, variance scaling of temperature and local intensity scaling (LOCI) of
precipitation (Teutschbein and Seibert, 2012). The bias-corrected precipitation and
temperature were calculated by using the equal weighted average method to obtain the
multi-model ensemble average values under the SSP2-4.5 and SSP5-8.5 scenarios,
which reduce the uncertainty caused by a single bias correction method and a single
GCM.
Table 1. Details of data from eight GCMs used in this study

| GCM | Institutions | Grid | Lon. × Lat. |
|---|---|---|---|
| ACCESS-CM2 | Australian Community Climate and Earth System Simulator | 192×144 | 1.875°×1.250° |
| ACCESS-ESM1-5 | Australian Community Climate and Earth System Simulator | 192×144 | 1.875°×1.250° |



| BCC-ECM1 | Beijing climate center | 320×160 | 1.125°×1.125° |
|---|---|---|---|
| CMCC-CM2-SR5 | Fondazione Centro Euro-Mediterraneo sui Cambiamenti Climatici | 288×192 | 1.25°×0.938° |
| CMCC-ESM2 | Fondazione Centro Euro-Mediterraneo sui Cambiamenti Climatici, | 288×192 | 1.25°×0.938° |
| GFDL-CM4 | National Oceanic and Atmospheric Administration | 144×90 | 2.5°×2° |
| MPI-ESM1-2-LR | Max Planck Institute for Meteorology | 192×96 | 1.875°×1.875° |
| NESM3 | Nanjing University of Information Science and Technology | 192×96 | 1.875°×1.875° |

**3.Methodology**

The catchment area was divided into 37 elevation bands ranging from 2960 m to 4820 m, with an interval of 50 m. These elevation bands were classified based on four landscapes: glacier, alpine desert, vegetation hillslope, and riparian zone. As a result, there were a total of 148 Hydrologic Response Units (HRUs) in the catchment. The landscape of alpine desert was the most widespread, covering an elevation range of 3425 m to 4727 m. The glacier was found in higher altitude areas, specifically between the elevation bands of 3725 m and 4727 m.

The model parameters used in this study were obtained from a previous study conducted in this catchment (Gao et al., 2022). These parameters are listed in Table 2.

The $\Delta h$-parameterization method was employed, which relies on empirical curves that are dependent on the size of the glacier. The study categorized glaciers into three size classes: large glacier, area > 20 km$^2$; medium-sized glacier, 5 km$^2$< area < 20 km$^2$; small glacier, area < 5 km$^2$. Both Glacier1 and Glacier2 had areas less



than 5 km2, making them small glaciers. The glacier mass balance (GMB) was
calculated using the glacier module of the FLEX-Cryo model. The calculated GMB
was then distributed to different elevation bands using the Δh-parameterization
method. Simultaneously, the glacier area and thickness were updated accordingly.
When a glacier was completely melted, the corresponding HRU was transformed into
alpine desert. The evolution of these landscapes was updated every 5 years (Wei et al.,

2023).

This study focused on the degradation of glaciers, changes in snow cover and

permafrost, and their impacts on runoff under climate warming. Factors such as solar
radiation, land surface temperatures influenced by snow cover, and vegetation
restoration were not considered. Average annual temperatures and annual precipitation
were used as indicators of future climate change. Glacier thickness at the highest
elevation band and glacier volume were used to quantify the thinning process of
glaciers. The maximum freeze depth of seasonal permafrost, thickness of the active
layer, and freeze-thaw cycle were used to characterize the thawing of frozen soil.
Snow cover days and snow water equivalent were utilized to measure the decreasing
trend of snow. Changes in runoff and runoff coefficient were analyzed to assess the
impact of mountain cryosphere degradation on water resources. Additionally, the
study examined the effect of degradation on runoff yield by observing the low runoff
during the early thaw season and the discontinuation of baseflow recession.

Table 2. Model parameters and their values or ranges in this study

| Parameter | Name | Prior range |
| --- | --- | --- |
| $F_{dd}$ (mm℃$^{-1}$d$^{-1}$) | Snow degree day factor | (1.0-5.0) |



| $C_g$ (-) | Glacier degree factor multiplier | (1.0-3.0) |
| --- | --- | --- |
| $S_{Umax\_V}$ (mm) | Root zone storage in vegetation hillslope | (50-200) |
| $S_{Umax\_D}$ (mm) | Root zone storage in alpine desert | (10-100) |
| $S_{Umax\_R}$ (mm) | Root zone storage in riparian wetland | (10-100) |
| $\beta$ (-) | The shape of storage capacity curve | (0-1) |
| $C_e$ (-) | Soil moisture threshold for reduction of evaporation | (0.1-0.6) |
| $D$ (-) | Splitter to fast and slow response reservoirs | 0.2 |
| $T_{lagf}$ (days) | Lag time from rainfall to peak flow | (0.8-3) |
| $K_f$ (days) | Fast recession coefficient | (1-10) |
| $K_s$ (days) | Slow recession coefficient | (10-100) |
| $k$ (W (m K)$^{-1}$) | Thermal conductivity | 2 |
| $\omega$ (-) | Water content as a decimal fraction of the dry soil weight | 0.12 |
| $\rho$ (kg/m$^3$) | Bulk density of the soil | 1000 |
| Pcalt | Precipitation increasing rate | 4.2 |
| Tcalt | Temperature lapse rate | 0.68 |

## 3.1 FLEX-Cryo model

FLEX-Cryo model is a landscape-based cryospheric hydrological model, which
considers multi-elements of cryosphere and their impacts on hydrology, including
glacier, snow and frozen soil. The elevation is also an important factor affecting the
temperature and precipitation. The temperature and precipitation are interpolated
based on the band in situ observation (2980 m). the temperature regression rate is -
0.68℃/100m and the precipitation increasing rate is 4.2%/100m. The value of 0 ℃ is
the threshold temperature to split snowfall ($P_s$) and rainfall ($P_l$).





### 3.1.1 Glacier and snow module


Glacier and snow melt were both calculated by the temperature - index method
which is on basis of the degree-day factor $F_{dd}$ (mm $\text{℃}^{-1}$ $d^{-1}$) ($F_{dd}$ in Table 2; equation
(11) in Table 3). Due to the lower albedo, the degree-day of ice is greater than snow,
and multiplied by a coefficient $C_g$ ($C_g$ in Table 2; equation (9) in Table 3). The glacier
area runoff $Q_g$ is calculated through the linear reservoir $S_g$ which the liquid rain $P_l$ and
glacier melt $M_g$ inflow and the runoff outflow ((Equation (4) in Table 3) and a
recession parameter $K_{fg}$ (Equation (1) in Table 3).

### 3.1.2 Frozen soil module


The Stefan equation was calculated at the different elevation bands based on the
interpolated temperature (Equation (18) in Table 3). The observed temperature was
multiplied by 0.6 to translate the air temperature to ground temperature which was
required in the equation. In this equation, the ε is the freeze / thaw depth (m), k is the
thermal conductivity (2 W $(\text{m K})^{-1}$), F is the freeze / thaw index(℃) which represents
the cumulative value of the temperature below (above ) $0\,℃$, $Q_L$ is the volumetric
latent heat of soil (J $m^{-3}$), L is the latent heat of the fusion of ice ($3.35 \times 10^5 J\ kg^{-1}$),
ω is the water content as a decimal fraction of the dry soil weight (0.12), and ρ is the
bulk density of the soil (1000 kg $m^{-3}$).
The frozen soil impacts on the runoff by the soil water and groundwater. In the
frozen season, the baseflow comes merely from the groundwater discharge at the
supra-permafrost layer (Qs). In the freezing season, when the freeze depth is greater
than 3 m and the supra-permafrost groundwater is frozen. And due to certain amount
of unfrozen water in frozen soil, the volume of slow reservoir $S_s$ (Equation (19) and
(20)) will be reduced to 10%. The groundwater system in seasonal frozen soil region





is still connected in winter. When the soil completely thaws at the lowest elevation
band, the runoff generated by the frozen $S_s$ will rapidly release to the $Q_s$. The
baseflow generation is assumed to be a linear recession process. The generated
baseflow is controlled by the reservoir $S_s$, recession coefficient $K_s$, the time t and
initial runoff $Q_0$ (Equation (21)).
$$\frac{dS_s}{dt} = R_s - Q_s - F_s \quad (19)$$

$$F_s = \begin{cases} 0.9 \cdot S_s & \varepsilon \geq 3m \\ -0.9 \cdot S_s & completely\ thaw \end{cases} \quad (20)$$

$$Q = Q_0 \cdot e^{-t/K_s} \quad (21)$$

**3.1.3 Rainfall-runoff module**

The root zone reservoir $S_u$ (equation (6) in Table 3), fast response reservoir $S_f$

(equation (7) in Table 3) and slow reservoir $S_s$ (equation (8) in Table 3) are critical
reservoirs for simulating rainfall-runoff processes. The runoff yield process is
governed by the root storage capacity and the water input to the soil (equation (12) in
Table 3) meanwhile the actual evaporation is determined by soil moisture and
potential evaporation. The generation runoff flows into two linear reservoirs ($S_f$ and
$S_s$) which represents the storm flow ($Q_f$) and groundwater runoff ($Q_s$), respectively.
The runoff yield process has similarity in alpine desert, vegetation hillslope and
riparian zone and the difference is the root zone storage capacity ($S_{umax}$). In the
vegetation hillslope, plants have well-developed root systems and the root zone has a
larger storage capacity. So, the $S_{Umax-V}$ was set with larger value. For the alpine desert
and riparian zone, the $S_{Umax-D}$ and $S_{Umax-R}$ were both limited due to the less developed
root system and storage capacity.



Table 3. The FLEX-Cryo model equations

| Landscape | Runoff equation | Water balance equation | Structural equation |
|---|---|---|---|
| Glacier | $Q_g = \dfrac{S_g}{K_{fg}}$ (1) | $\dfrac{dS_g}{dt} = P_l + M_g - Q_g$ (4) | $M_g = \begin{cases} F_{dd} \cdot T \cdot C_g & S_W = 0 \text{ and } T > 0 \\ 0 & S_W > 0 \text{ and } T \le 0 \end{cases}$ (9) <br><br> $\Delta h = (h_r - 0.30)^2 + 0.60(h_r - 0.30) + 0.09$ (10) |
| Alpine desert <br><br> Hillslope vegetation <br><br> Riparian area | $Q_f = \dfrac{S_f}{K_f}$ (2) <br><br><br> $Q_s = \dfrac{S_s}{K_s}$ (3) | $\dfrac{dS_W}{dt} = P - M_W$ (5) <br><br> $\dfrac{dS_u}{dt} = P_l + M_W - E_a - R_u$ (6) <br><br> $\dfrac{dS_f}{dt} = R_f - Q_f$ (7) <br><br> $\dfrac{dS_s}{dt} = R_s - Q_s$ (8) | $M_W = \begin{cases} F_{dd} \cdot T & T > 0 \\ 0 & T \le 0 \end{cases}$ (11) <br><br> $R_U = (P_l + M_W) \cdot \left( 1 - \left( 1 - \dfrac{S_U}{S_{U\max}} \right)^\beta \right)$ (12) <br><br> $E_a = E_p \cdot \left( \dfrac{S_U}{C_e \cdot S_{U\max}} \right)$ (13) <br><br> $R_f = R_U \cdot D$ (14) <br><br> $R_s = R_U \cdot (1-D)$ (15) <br><br> $R_{fl}(t) = \sum\limits_{i=1}^{T_{lagf}} cf(i) \cdot R_f(t-i+1)$ (16) <br><br> $c_f(i) = \dfrac{i}{\sum\limits_{u=1}^{T_{lagf}} u}$ (17) <br><br> $\varepsilon = \left( \dfrac{2 \cdot 86400 \cdot k \cdot F}{Q_L} \right)^{0.5} = \left( \dfrac{2 \cdot 86400 \cdot k \cdot F}{L \cdot \omega \cdot \rho} \right)^{0.5}$ (18) |


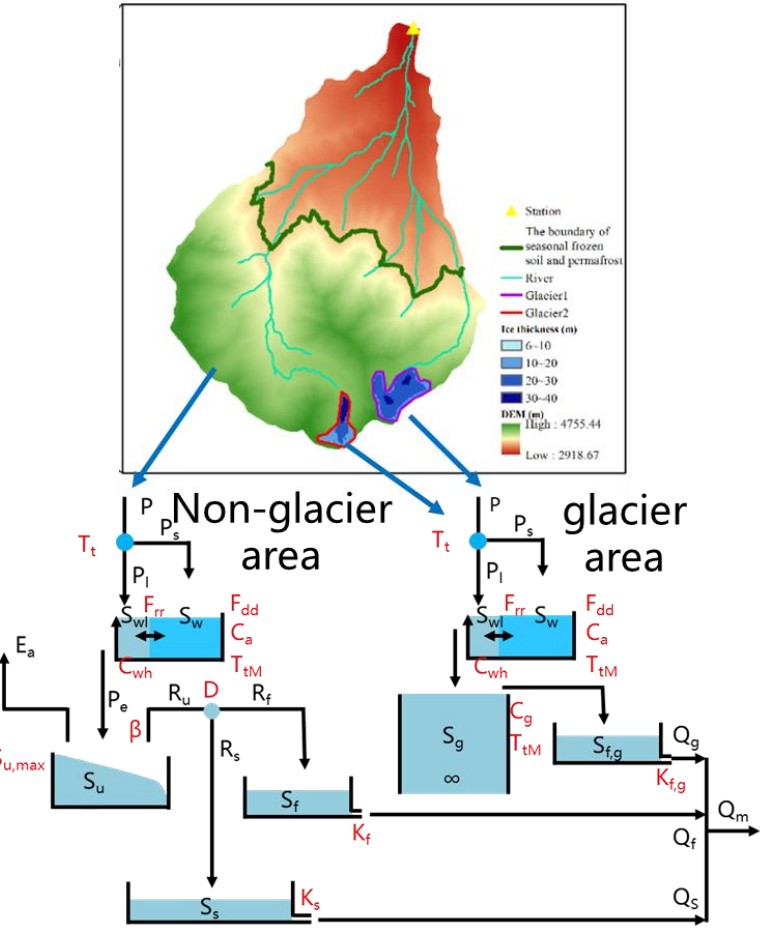


Figure.3 Structure of the FLEX-Cryo model. The abbreviation in red color

indicates paraments and the abbreviations in black indicate storage components and
fluxes.
**4. Results**
**4.1 Future climate change**

Figure 4 shows the prediction of future climate in 2015-2100 under the SSP2-4.5

and SSP5-8.5 scenarios based on the average values of eight climate models (adjusted
for bias). According to the SSP2-4.5 scenario, the temperature will increase by 2.07°C
relatively steadily by 2100. Under the SSP5-8.5 scenario, temperatures are projected





to continue to rise by 5.04°C over the course of the century. Precipitation changes are
more drastic than temperature, especially after the eighties of the 21st century under
the SSP5-8.5 scenario. Overall, the precipitation under the SSP2-4.5 scenario
increased by 14.25 %, and the precipitation increased by 33.50 % under the SSP5-8.5
scenario. Before the 80s of the 21st century, the increase in precipitation was almost
the same under different scenarios, about 8.9 mm 10 years$^{-1}$ and 8.5 mm 10 years$^{-1}$,
respectively.

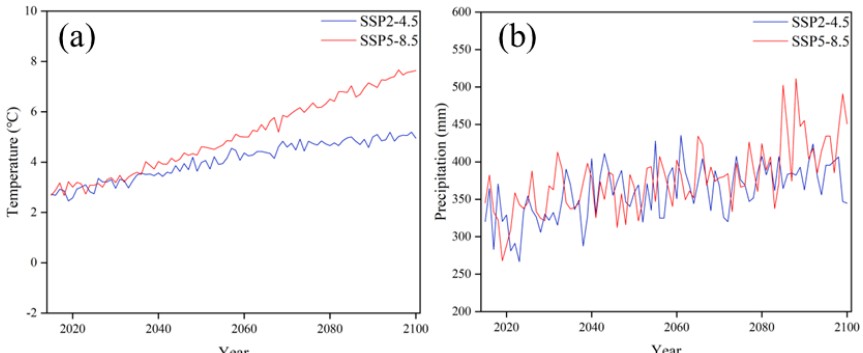

Figure 4. The annual average temperature (a) and annual precipitation mean (b) of
bias adjusted multi-Global Climate Model from 2015-2100.
**4.2 Predicting glacier retreat**

In the initial status (Figure 5), the Glacier1 and Glacier2 had areas of 8.78×10$^5$

m$^2$ and 4.08×10$^5$ m$^2$, and ice volumes of 20.13×10$^6$ m$^3$ and 8.86×10$^6$ m$^3$, respectively.
Glacier1 exhibited greater thickness and volume compared to Glacier2.

Both Glacier1 and Glacier2 experienced retreat, characterized by a decrease in

glacier volume and thinning of glacier thickness (Figure 5). Starting from the 2020s,
the glacier volume showed a rapid decline, and after the 2030s, the highest-altitude
portion of the glacier entered a phase of rapid thinning. Around 2040, the glacier
degradation reached a stabilization period, during which glaciers were only present in



the highest elevation band. According to the SSP2-4.5 scenario, Glacier1 and Glacier2
are projected to completely melt and disappear by 2051 and 2046, respectively. Under
the SSP5-8.5 scenario, the complete melt-out time is slightly earlier, occurring in
2045 and 2044 for Glacier1 and Glacier2, respectively. After the glaciers completely
melt, approximately 5.6% of the ablated glacier area will transform into alpine desert.
Taking the glacier changes in 2025, 2035, and 2045 as examples, under the
SSP2-4.5 scenario, the area of Glacier1 is projected to decrease to $5.49\times10^5$ m$^2$,
$1.52\times10^5$ m$^2$, and $0.26\times10^5$ m$^2$, with corresponding volume reductions to $5.27\times10^6$
m$^3$, $1.03\times10^6$ m$^3$, and $0.26\times10^6$ m$^3$, respectively (Figure 7). Comparatively, the retreat
trend is more pronounced under the SSP5-8.5 scenario. The area of Glacier1 is
projected to be $4.00\times10^5$ m$^2$, $0.81\times10^5$ m$^2$, and $0.26\times10^5$ m$^2$, with volumes of
$4.86\times10^6$ m$^2$, $0.71\times10^6$ m$^3$, and $0.03\times10^6$ m$^3$, respectively. The degradation of
Glacier2 follows a similar pattern to that of Glacier1, except that Glacier2 experiences
less ice loss. According to the SSP5-8.5 scenario, Glacier2 is projected to completely
melt by 2045. In 2025 and 2035, the area of Glacier2 remains consistent, with values
of $1.67\times10^5$ m$^2$ and $0.51\times10^5$ m$^2$ for both scenarios, respectively. These glaciers are
only distributed within the elevation bands from 4625 m to 4727 m and from 4675 m
to 4727 m.

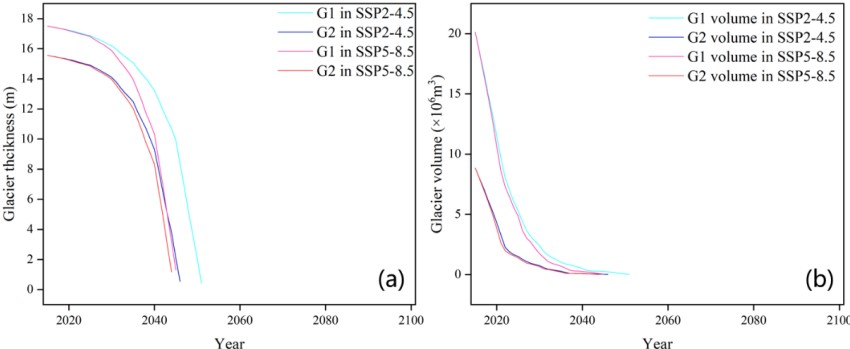




346   Figure 5. The glacier thickness (a) and glacier volume (b) change from 2015 to

347  2100 for the Glacier1 and Glacier 2

348  **4.3 Forecasting the degradation of frozen soil**

349   In the initial state (Figure 6), the seasonal frozen soil exhibited an early freeze

350  onset in early November, with a freeze duration of approximately 200 days at the

351  lowest elevation band. The permafrost, on the other hand, experienced a thaw onset in

352  mid-June, lasting around 120 days at the highest elevation band. The maximum freeze

353  depth was approximately 2.30 m at lower altitudes, while the active layer thickness

354  measured around 1.27 m at the highest elevation.

355   By the end of the 21st century, under the SSP2-4.5 scenario, several changes are

356  projected to occur. The freeze onset of seasonal frozen soil will be delayed by 10

357  days, resulting in a shortened freeze-thaw cycle duration of approximately 1 month.

358  The thaw onset of permafrost will be advanced by 19 days, leading to an increased

359  freeze-thaw cycle duration of nearly 50 days. Additionally, the maximum freeze depth

360  is expected to decrease by 5.17 cm per decade, while the active layer thickness will

361  increase by approximately 8.24 cm per decade. The degradation trend of permafrost is

362  more severe under the SSP5-8.5 scenario. By the end of the 21st century, compared to

363  the SSP2-4.5 scenario, the freeze onset of seasonal frozen soil will be shortened by 22

364  days, resulting in a further reduction of the freeze-thaw cycle duration by over 2

365  months. The thaw onset of permafrost will occur approximately 1 month earlier, and

366  the freeze-thaw cycle duration of permafrost will increase by nearly 3 months. The

367  decreasing trend of the maximum freeze depth and the increasing trend of the active

368  layer thickness are approximately twice as pronounced under the SSP5-8.5 scenario

369  compared to the SSP2-4.5 scenario. While the freeze onset of seasonal frozen soil





exhibits significant variation between consecutive years, the other frozen soil
elements follow a more stable change pattern. Before 2040, there is little difference
between the two scenarios, except for the active layer thickness. Seasonal frozen soil
will begin to freeze around mid-November and late November, while permafrost will
start to thaw in mid-May and early June by the year 2100.

Under the SSP2-4.5 and SSP5-8.5 scenarios, the lower limit of permafrost

gradually expands along the altitudinal gradient, with rates of 4.30 m per year and
8.75 m per year, respectively (Figure 7). In the SSP2-4.5 scenario, the lower limit of
permafrost is projected to reach altitudes of 3685 m, 3795 m, 3835 m, 3865 m, 3985
m, and 4015 m in the years 2025, 2035, 2045, 2055, 2075, and 2095, respectively. The
lower limit of permafrost in 2095 under the SSP2-4.5 scenario is comparable to the
lower limit of permafrost (3965 m) in 2055 under the SSP5-8.5 scenario. Before 2045,
the lower bound exhibits similar changes under both scenarios, but a significant
divergence occurs afterward. The lower limit is projected to increase to 4355 m by
2095 under the SSP5-8.5 scenario.



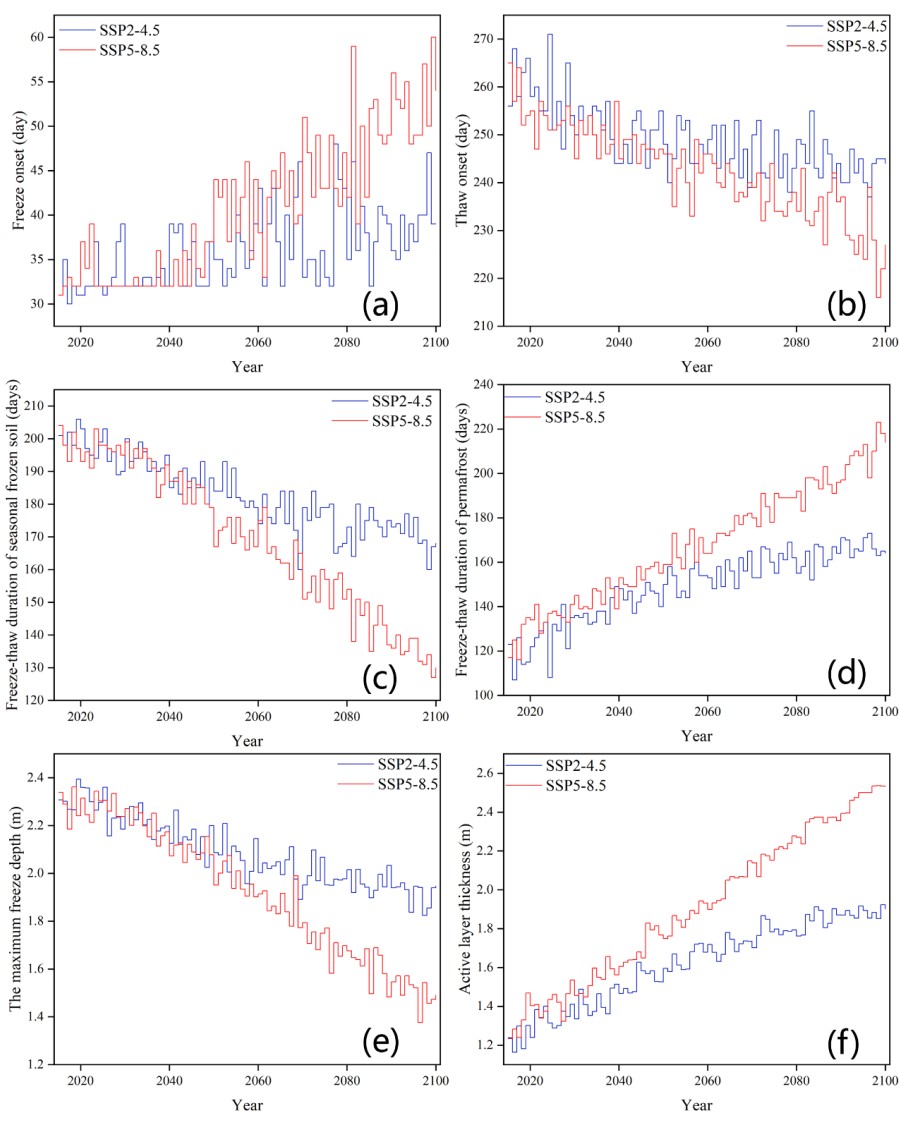

Figure 6. Changes in seasonal frozen soil and permafrost from 2015-2100 under SSP2-4.5 and SSP5-8.5 scenarios. (a, b) Freeze and thaw onset. (c, d) Freeze-Thaw duration of frozen soil and permafrost. (e, f) The maximum freezing depth and active layer thickness.



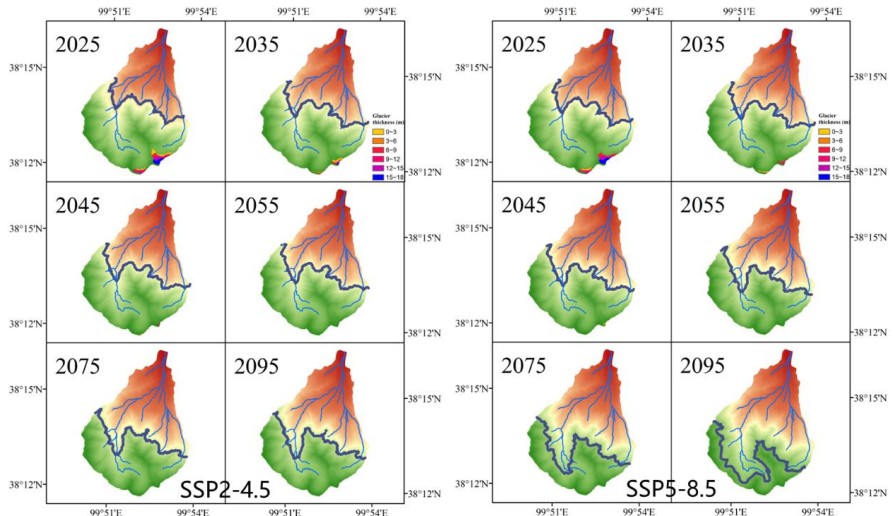


Figure 7. Changes of ice thickness and the lower limit of permafrost in 2025, 2035,
2045, 2055, 2075 and 2095 under SSP2-4.5 and SSP5-8.5.
**4.4 Snow change in the future**

The duration of snow cover is projected to decrease continuously in the future.

Under the SSP2-4.5 scenario, the snow cover days are likely to be shortened by 45
days, while under the more severe SSP5-8.5 scenario, the reduction is expected to be
around 76 days. Simultaneously, the snow water equivalent, which measures the
amount of water contained in the snowpack, is projected to exhibit more variable
changes but with an overall decreasing trend. For the SSP2-4.5 scenario, the snow
water equivalent will decrease by 0.24 mm per year, resulting in a reduction of
approximately 41.4%. Under the SSP5-8.5 scenario, the decrease in snow water
equivalent is more pronounced, with a drop of 0.35 mm per year, corresponding to a
reduction of up to 46.0%.





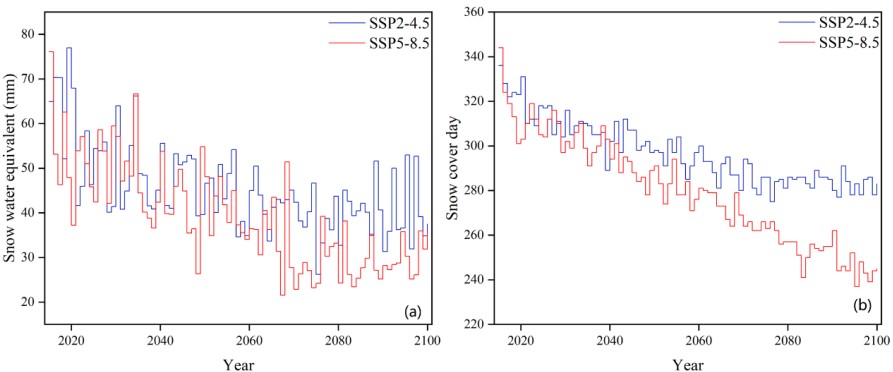


Figure 8. (a) snow water equivalent at entirely watershed and (b) snow cover day at

the highest band.

**4.5 Project future runoff**

The depth of runoff in the entire basin shows a declining trend in the future.

Under both the SSP2-4.5 and SSP5-8.5 scenarios, before the complete melting of

glaciers, the runoff depth is estimated to be around 257 mm and 277 mm, respectively.

After the glaciers completely melt, the runoff depth is projected to decrease by

15.56% and 18.05% for the SSP2-4.5 and SSP5-8.5 scenarios, respectively (Figure 9).

By 2100, the average annual runoff depth is expected to be similar for both scenarios,

at approximately 217 mm and 227 mm. In the SSP2-4.5 and SSP5-8.5 scenarios, the

tipping point for the runoff depth in the glacier area is projected to occur in 2021 and

2019, respectively, with values of 155.93 mm and 175.98 mm. After reaching the

tipping point, the runoff depth in the glacier area is likely to continue decreasing until

the glaciers completely melt. In non-glacier areas, the runoff depth shows an increase

of 0.48 mm per year and 0.65 mm per year.

The runoff coefficient, which represents the proportion of precipitation that

becomes runoff, follows a similar pattern to the glacier runoff changes. It initially

increases, then decreases, and eventually reaches a relatively stable state after the



glaciers completely melt (Figure 10). The maximum values of the runoff coefficient
occur in 2021 and 2019, coinciding with the tipping points of the glacier runoff. By
the end of the 21st century, the runoff coefficient is projected to be dramatically
reduced to approximately 0.42.

Two hydrological phenomena observed in permafrost mountainous catchments,

namely the low runoff in the early thawing season (LRET) and discontinuous
baseflow recession (DBR) (Gao et al., 2022), are expected to persist in the future
(Figure 11). Meanwhile, baseflow, which represents the sustained flow of water from
groundwater, shows an increasing trend. The duration of the early thawing season is
projected to be further reduced. The first recession coefficient remains unchanged,
while the second recession coefficient progressively increases. Under the SSP2-4.5
scenario, the second recession coefficient is equal to 74 days, which is consistent with
the recession coefficient in 2060 under the SSP5-8.5 scenario. This suggests that the
permafrost area undergoes less significant changes under SSP2-4.5 scenario than
SSP2-8.5 scenario according to Figure 7. The baseflow gradually increases, especially
in the SSP5-8.5 scenario, as indicated by the runoff depth on a logarithmic scale.

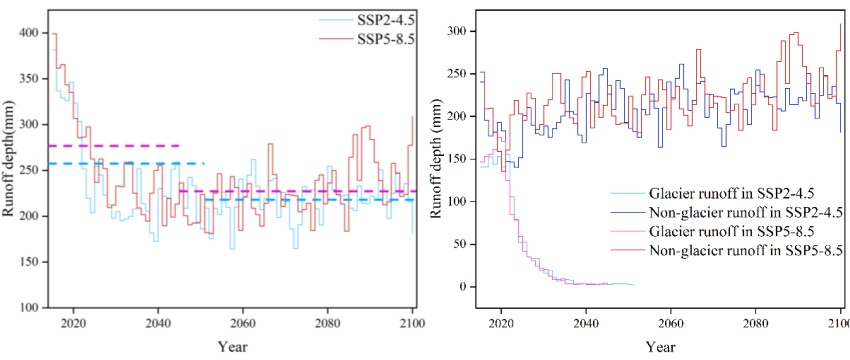

Figure 9. (a) The predicted runoff depth of the total basin (b) Runoff in the glacier and
in the non-glacier from 2015-2100



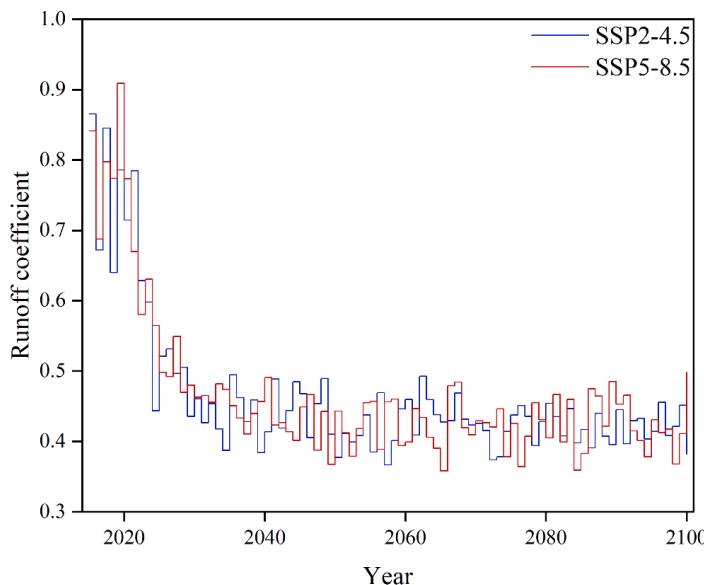


Figure 10. Project runoff coefficient under SSP2-4.5 and SSP5-8.5 scenarios.

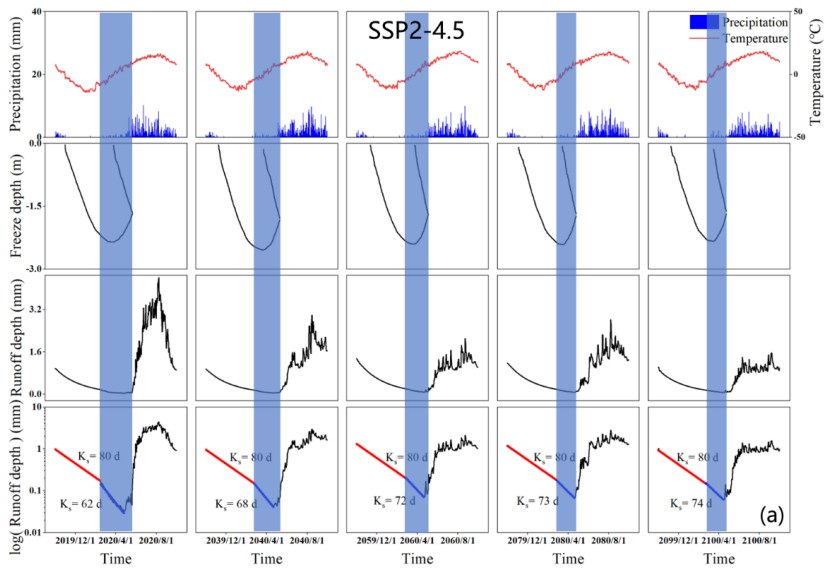






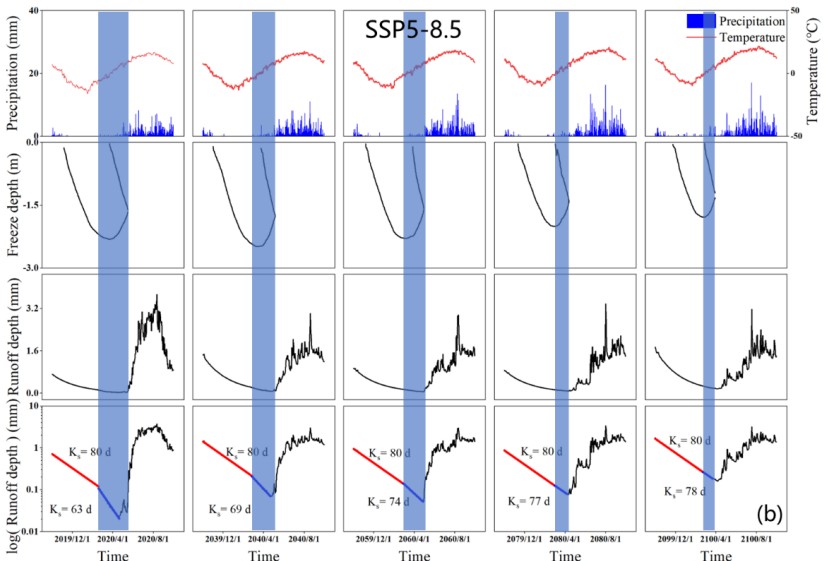

Figure 11. Temperature, precipitation, runoff depth and freeze-thaw cycle in 2020,

2040, 2060, 2080 and 2100 under SSP2-4.5 (a) and SSP5-8.5 scenarios (b).

**5.Discussion**

**5.1 Changes of the mountain cryosphere in future**

The cryosphere, which encompasses glaciers, snow, frozen soil, and permafrost,

plays a vital role in storing approximately 75% of the world's freshwater resources,

while around 17% of the global population resides in cryosphere regions (Qin et al.,

2021). Understanding the changes occurring in the cryosphere is crucial for assessing

the long-term sustainability of water resources (Whitfield et al., 2021). The Hulu

catchment, located in the northeast Tibet Plateau, exhibits a diverse distribution of

cryosphere elements, making it an ideal area for studying these changes (Gao et al.,

2019; Xu et al., 2019). However, there is a lack of research on the degradation of

multiple cryosphere elements within the Hulu catchment and its implications for



future hydrology, despite the Heihe River Basin being recognized as a typical region
for studying hydrological and water resource changes in cold regions (Ning et al.,
2008). In this study, we projected a future warming trend of 0.3°C per decade and
0.6°C per decade, accompanied by an increase in precipitation of 7.9 mm per decade
and 12.0 mm per decade under the SSP2-4.5 and SSP5-8.5 scenarios, respectively
(Figure 4). These projections align with the findings of Chen et al. (2022), who
observed similar warming trends of 0.3-0.4 °C per decade and 0.7-0.8°C per decade,
as well as precipitation increases ranging from 1.6-14.8 mm per decade and 6.0-20.6
mm per decade under similar scenarios. This consistency between our projections and
previous research supports the reliability of the forcing data used in the FLEX-Cryo
model.

Furthermore, Wang et al. (2018) predicted an increase in the annual maximum

freeze depth in the Heihe River Basin from 2011 to 2066 at a rate of 5.4 cm per
decade, which closely aligns with the predicted change of 5.2 cm per decade in our
study (Figure 6). While there have been limited studies investigating future changes in
glaciers and other cryosphere elements within the Hulu catchment, including the Shiyi
Glacier, conducting a comparative analysis with projections from other regions can
enhance our understanding of mountain cryosphere retreat. Although the comparative
analysis approach may have some limitations in terms of rigor, it provides valuable
insights (Han et al., 2023).
**5.2 The effect of the mountain cryosphere degradation on runoff**

Glaciers and snow cover play a crucial role in water retention, with meltwater

contributing significantly to downstream water resources and the ecological
environment (Stecher et al., 2023). The turning point of glacier runoff represents a



critical tipping point that signifies not only the thinning of glaciers but also the
irreversible stage of water resources in the basin (Brovkin et al., 2021). In the Hulu
catchment, the proportion of glacier runoff reached 51% to 55% between 2019 and
2021, indicating that it is in the turning point period (Figure 9). Subsequently, the
contribution of glacier runoff gradually decreases until complete melting occurs.
Temperature is the primary factor influencing glacier runoff, while precipitation and
temperature together determine the proportion of glacier runoff in relation to total
runoff. Although the highest contribution of glacier runoff and the tipping point of
glacier runoff may not align precisely, after the tipping point, the capacity of glacier
runoff to contribute to overall runoff continuously diminishes. From 2015 to 2021,
there has been a decreasing trend in precipitation, leading to a corresponding decline
in non-glacier runoff (Figure 4 and 9). Thus, while glacier runoff has increased, the
total runoff has decreased. However, between 2032 and 2038, even though rainfall
continues to decline, the contribution of glacier runoff to overall runoff becomes
negligible due to the limited volume of ice remaining (glacier volume < $1 \times 10^6$ m³),
resulting in minimal glacier melting runoff (Figure 5 and Figure 9). On the other
hand, once the glaciers have completely melted, the total runoff in the Hulu catchment
is reduced by 16% to 18%, and the runoff coefficient is halved (Figure 9 and Figure
10). This highlights the critical role of glaciers as solid freshwater reservoirs in
regulating water sources and mitigating droughts (McCarthy et al., 2022).

The freeze-thaw cycle has a significant impact on runoff yield and hydrological

response routines in the Hulu catchment (Sun et al., 2022; Wang et al., 2020).
Precipitation in the Hulu catchment is primarily concentrated in the summer when soil
moisture is high and even close to saturation, making saturation excess flow the main




mechanism for runoff generation (Li et al., 2016). During the freeze-thaw cycle, the
weak permeability of frozen soil affects both surface runoff and infiltration. Soil
runoff primarily occurs through underground in hillslope and surface water flow in
riparian area, resulting in a faster response to rainfall and snowmelt and contributing
to a higher runoff coefficient (Hu et al., 2022; Jones et al., 2023). However, it is
important to note that shallow frozen soil does not completely block the interaction
between deeper soil layers and the surface. Frost heave in the soil creates large pores,
allowing snowmelt water and precipitation to bypass the matrix layer and reach the
deeper soils (Jiang et al., 2021; Zhang et al., 2023). This phenomenon is considered
one of the significant reasons for low runoff in the early thawing season (Mohammed
et al., 2021). Low runoff is observed between the frozen season and complete thawing
season (Figure 11). The duration of freeze-thaw cycles in seasonal frozen soils is
shortening, and freeze onset is being delayed due to the warming climate, resulting in
a decreasing duration of low runoff. However, the temperature during the freezing
season remains lower than the initial frost heave temperature of the soil, and there is
still a deficit of soil water in the early thaw, indicating that the prevalence of low
runoff will persist in the future (Teng et al., 2022; Wen et al., 2024).

The freezing state has a significant impact on the recession process of baseflow,

and permafrost plays a crucial role in discontinuous baseflow (Cooper et al., 2023; J.
Wang et al., 2022). During the freezing season, baseflow follows a linear recession
process ($K_s$ = 80 days), with contributions from both permafrost and seasonal frozen
soil regions (Figure 11). In the frozen season, the groundwater under the supra-
permafrost layer becomes inactive, and baseflow is solely derived from the seasonal
frozen soil regions, causing a discontinuous recession. With climate warming, the



lower limit of permafrost gradually moves upward along the elevation, resulting in the
shrinking of the permafrost region. This suggests that in the future, an increased
proportion of baseflow will originate from the expanding area of seasonal frozen soil,
leading to a gradual decrease in the influence of permafrost on baseflow.
Consequently, the discontinuous recession of baseflow will gradually transition into a
linear recession. Furthermore, an increase in the thickness of the active layer enhances
the soil water storage capacity, contributing to a gradual rise in baseflow (Yao et al.,

2021).

**5.3 Uncertainty and limitations**
The uncertainty in this study arises from the forcing data of the General
Circulation Models (GCMs), the bias correction methods, and the parameters selected
in the FLEX-Cryo model (Wilby and Harris, 2006). The coarse spatial resolution of
the GCMs prevents a comprehensive description of the climate at the basin scale,
particularly in plateau and mountainous regions heavily influenced by altitude. The
selection of parameters for the FLEX-Cryo model is also a significant source of
uncertainties. Due to the complex topography in the mountain cryosphere, degree-day
factor, altitude effect on climate and soil water storage capacity cannot be fully
reflected at the catchment scale. To mitigate some of the uncertainties associated with
the GCM outputs, a multi-model and multi-method approach is employed in this
study. The equal weighted average method is used to combine the values from
different models and methods, aiming to reduce uncertainties and provide a more
robust assessment of the results. It is important to note that the optimal parameter
group selected for the FLEX-Cryo model in this study has been chosen based on
previous research (Gao et al., 2022). While this helps to establish a more reliable




parameterization, there may still be inherent limitations in the chosen parameter
values. Overall, the uncertainties and limitations associated with the forcing data, bias
correction methods, and parameter selection in the FLEX-Cryo model need to be
considered when interpreting the results of this study. Further research and
improvements in these areas can enhance the accuracy and reliability of future
assessments of the effects of mountain cryosphere degradation on runoff.
**6.Conclusions**

The mountain cryosphere, encompassing glaciers, snow, and frozen soil, plays a

critical role in downstream water resources and the ecological environment.
Understanding its response to climate change is crucial for effective water resource
management and flood prevention. In this study, we employed the FLEX-Cryo model
and data from eight Global Climate Models (GCMs) under the SSP2-4.5 and SSP5-
8.5 scenarios to project the potential impacts of climate change on the mountain
cryosphere and hydrology. Based on our simulation results, the following conclusions
can be drawn:

(1) The air temperature is projected to increase by 2.1 ℃ and 5 ℃ by 2100,

while precipitation is expected to increase by 8 mm/10 years and 12 mm/10 years.
These changes in temperature and precipitation patterns indicate a significant shift in
the climatic conditions of the study area.
(2) Glacier and snow cover are anticipated to experience retreat and shrinkage in the
future. Under the SSP5-8.5 and SSP2-4.5 scenarios, glaciers are projected to
completely melt by 2045 and 2051, respectively. Additionally, the duration of snow
cover will be shortened by 45 days and 76 days, while the snow water equivalent will
decrease by 0.24 mm/yr and 0.35 mm/yr.



(3) The frozen soil is expected to undergo degradation. By 2100, the freeze onset of
seasonal frozen soil is projected to delay by 10 days and 22 days, and the thaw onset
of permafrost is expected to advance by 19 days and 32 days. The lower limit of
permafrost is estimated to reach altitudes of 4015 m and 4355 m along the altitudinal
gradient. Moreover, the maximum freeze depth will decrease by approximately 5.17
cm/10 years and 10.93 cm/10 years, while the active layer thickness will increase by
8.24 cm/10 years and 15.47 cm/10 years.
(4) The degradation of the mountain cryosphere has significant implications for water
resources in the catchment area, particularly in terms of runoff yield. The tipping
point for glacier runoff occurred between 2019 and 2021. Once the glaciers have
completely melted, the depth of runoff is projected to decrease by approximately 16%
and 18%. However, in non-glacier areas, the depth of runoff is expected to increase by
0.22 mm/yr and 1.07 mm/yr from 2015 to 2100. By the end of the 21st century, the
runoff coefficient in the catchment is projected to reach approximately 0.42.
Importantly, the duration of low runoff during the early thawing season will be
shorter. The discontinuous recession of baseflow is gradually transitioning towards a
linear pattern, resulting in increased baseflow. The second recession coefficients are
estimated to be around 74 days and 78 days, respectively, by the year 2100.
In conclusion, this study provides insights into the potential impacts of climate change
on the mountain cryosphere and hydrology. The projected changes in temperature,
precipitation, glacier retreat, snow cover, and frozen soil dynamics highlight the
urgent need for proactive water resource management strategies in the face of a
changing climate. Further modelling research and monitoring efforts are necessary to
refine these projections and guide effective adaptation measures to sustainably



manage water resources in mountainous regions.

**Competing interests**
At least one of the (co-)authors is a member of the editorial board of Hydrology and
Earth System Sciences.

**Acknowledgements**
This research has been supported by the National Natural Science Foundation of
China (grant no. 42071081 and 42122002). Zheng Duan acknowledges the support
from the Crafoord Foundation (No. 20210552).

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
