# Peer review of "Projected future changes in cryosphere and hydrology of a mountainous"

_EGUsphere, 2023_

## Author Comment (AC1)

This manuscript conducted a systematic projection on the runoff and cryospheric elements including glacier, snow and frozen soil in a typical mountainous catchment. Overall, the manuscript is well structured and written and easy to follow. It is suitable for publication in HESS, especially for this special issue. However, I would like to point out two major concerns regarding the uncertainty and reliability of the results

Reply: We thank Referee #1 for positive remarks about the manuscript's structure and suitability for publication in HESS. Regarding the concerns about uncertainty and reliability of the results, we will address these issues in the following ways and will detail them further in the revised manuscript.

The FLEX-Cryo model is an extension of the FLEX-Topo-FS model by coupling the $\Delta$h-parameterization method to estimates the evolution of glacier. Firstly, similar to the FLEX-Topo-FS model, we followed the same top-down approach to construct the model: data analysis → qualitative perceptual model → quantitative conceptual model → testing of model realism, which means the model structure is consistent. The parameters were carefully selected through the Generalized Likelihood Uncertainty Estimation (GLUE) method and have been robustly validated in the study by Gao et al (2022) study. No new parameters were introduced in the FLEX-Cryo model compared to the FLEX-Topo-FS model, maintaining the reliability previously established by Gao et al. (2022). Therefore, the results of the FLEX-Cryo are reliable.

The Global Climate Models (GCMs) were selected based on the previous validated studies. Despite inherent regional simulation uncertainties associated with each GCM, we have applied widely used statistical downscaling, bias calibration, and equal-weighted average methods to mitigate these uncertainties. By using well-validated GCMs and the widely used procedures for refining their outputs, we have enhanced the reliability of the FLEX-Cryo model results.

In conclusion, we are confident that the model's calculations are reliable, and that we have effectively managed the associated uncertainties. These verification details will be incorporated into the revised manuscript.

1. The model validation is poorly conducted. Although the authors claimed that the parameters are adopted from a previous study in this catchment, some results related to model performance should be presented to show the confidence of model. If I understand correctly, the model in this study is the combination of the model in Gao et al. (2022) and the Δh-parameterization. Isn't there new parameter brought by this module compared to the previous version? Could the -Cryo model simulate something that cannot be simulated by -FS model, and if so, how does the model perform on simulating this additional objective? It is rather easy to simulate the relative change of glacier thickness, but simulating the absolute thickness of glacier is difficult, which significantly influences the conclusions such as the time glacier will disappear. So, again, please present some results to show reliability of glacier simulation. Even though

all the simulations are the same with -FS model, some results need to be provided to show the confidence of model.

Reply: Thank you for your constructive feedback. We acknowledge the importance of thoroughly validating the FLEX-Cryo model and will include additional verification in the revised manuscript.

As clarified above, the FLEX-Cryo model integrates the robust FLEX-Topo-FS framework with the Δh-parameterization method, enabling it to simulate not only glacier melt and runoff from glacial regions but also the evolution of glacier characteristics, including changes in glacier area and thickness, and their hydrological impacts. The 11 parameters deployed in this study were selected using GLUE and validated by Gao et al. (2022), while the remaining parameters were determined based on prior research and empirical measurements. Importantly, this model does not introduce new parameters.

To address your specific concerns, we will present additional results, including Figure 5(b), which depicts the prediction of actual glacier thickness changes at the highest elevation band. This figure represents absolute glacier thickness changes, rather than relative changes, providing a clearer insight into the model's capability to simulate critical glaciological variables that influence projected timelines for glacier disappearance.

2. The uncertainty issue is addressed inadequately, although the authors mention it in the limitation section. I understand that this study aims to perform a systematic projection on the mountain cryosphere and hydrology, thus does not discuss much about the uncertainties of model parameter and GCM bias correction. However, I think the authors should at least report the uncertainties from different GCMs, given that eight GCMs are adopted for climate projection. The uncertainty range should be provided for the values in the main text (e.g., L304~314) and Figures (e.g., Figure 4).

Reply: This is a very good suggestion. Indeed, each GCM in CMIP6 demonstrates varying simulation capacities across different regions, resulting in a spectrum of uncertainties. We will quantify and report these uncertainties from the eight GCMs used for climate projection to provide a more comprehensive understanding of the potential variability in our results. In the revised manuscript, we will include an uncertainty range for the values discussed in the main text and depicted in figures (e.g., Figure 4).

Other specific issues:

1. Please adjust the paragraph format to justified.

Reply: We will do this in the revised manuscript.

2. Please provide a table to list the meaning of all the variables in Table 3 and Figure 3, to make them easier to find.

Reply: We will do this in the revised manuscript.

3.  There are many GCMs in CMIP6. Why are these eight GCMs selected?

Reply: The eight GCMs are all models that have been well validated in the adjacent catchments by previous studies. We will add this reason for the selection of these eight GCMs in the revised manuscript.

I suggest the authors to reconstruct the Methodology section to make it more readable. It would be better to introduce the model first, and then introduce the spatial discretization of the catchment (the first paragraph of the current Methodology section), because the elevation band and HRU is the simulation unit of the model. Besides, more details of Δh-parameterization method need to be provided in the 3.1.1 section. The current description is too general, which is difficult to understand for a reader not familiar with this method.

Reply:This is a really good advice and we agree with you. The introduction of the model belongs to the construction of conceptual model and the spatial discretization is the part of procedural model. According to the construction principle of the hydrological model (Beven, 2012), the conceptual model building precedes procedural model. In our revised manuscript, we will adjust the structure of the Methodology section accordingly. The Δh-parameterization method is the key for the extension of the FLEX-Topo-FS model to FLEX-Cryo model and it is also the core module for calculating the glacier evolution. Detailed description of the Δh-parameterization method will be added in the revised manuscript.

4.  Why a single value is provided for some parameters in Table 2, but a range is provided for others?

Reply: This is a very good question. The parameters for FLEX-Cryo model are the selected optimal results based on the study Gao et al (2022) using GLUE method. The parameters with range value are the parameter uncertainty range which is also list in Gao et al (2022). The single value is determined based on the previous study and measurement. The parameters presented with a single value were determined based on previous study by Gao et al. (2022) and field measurements, reflecting the most suitable values for these parameters within our specific catchment context. The parameters provided as a range represent the uncertainty bounds identified in the study by Gao et al. (2022). To avoid confusions, we will add a new table to list the optimal parameters and add more necessary explanations in the revised manuscript.

Reference:

Beven    K.,    Rainfall-Runoff    Modelling:    The    Primer,    1–23, https://doi.org/10.1002/9781119951001.ch1, 2012.

Gao, H., Han, C., Chen, R., Feng, Z., Wang, K., Fenicia, F., and Savenije, H.: Frozen soil hydrological modeling for a mountainous catchment northeast of the Qinghai–Tibet Plateau, Hydrol. Earth Syst. Sci., 26, 4187–4208, https://doi.org/10.5194/hess-26-4187-2022, 2022.

---

## Author Response (AR1)

**Public justification (visible to the public if the article is accepted and published)**:
Dear Authors,

Please revise the manuscript based on all the comments raised by the three reviewers and your responses. In addition, based on my reading of the abstract, I have two following minor comments for your reference:

1) there is still a lack of systematic research that evaluates the variation of cryospheric elements in mountainous catchments and their impacts on future hydrology and water resources. I don't think this statement is reasonable. I acknowledge that such an issue has not been adequately addressed, but not lack.

Reply: We thank the Editor's efforts in handling our manuscript and providing us with constructive comments to improve its quality. Based on your comment, we have revised the sentence in the Abstract. The updated version reads as follows:

"These changes dramatically alter the local and downstream hydrological regime, posing significant threats to basin-scale water resource management and sustainable development. Despite the critical nature of this issue, it remains inadequately addressed, particularly in mountainous catchments."

To strengthen the background and context of our study, we have also added more relevant references in the Introduction and discussion section in our revised manuscript (see Line 500-502, and Line 130-133).

2) Regarding hydrology, runoff exhibits a decreasing trend until the complete melt-out of glaciers, resulting in a total runoff decrease of 15.6% and 46 18.1%, and the following several sentences about the results. I am confused by these sentences. Here you said the total runoff decreases, but why do you have two numbers, and in the following sentence you said the total runoff increases due to precipitation change. The results are not logically organized and clearly expresed. Also, what do you mean by 'runoff', 'total runoff'? Please rewrite/rephrase them to clarify the results.

Reply: Thank you for pointing out the confusions. To address this issue, we have revised the Abstract to better explain the runoff change under two scenarios (SSP2-4.5 and SSP5-8.5). The updated version of the Abstract now reads as follow:

"Regarding hydrology, catchment total runoff exhibits a decreasing trend, with the tipping point of glacier runoff occurring approximately between 2019 and 2021. Furthermore, permafrost degradation will likely reduce the duration of low runoff in the early thawing season, causing the discontinuous baseflow recession to gradually

transition into linear recessions and resulting in an increase in baseflow. Our results highlight the significant changes expected in the mountainous cryosphere and hydrology in the future."

**Responses to Anonymous Referee #1**

This manuscript conducted a systematic projection on the runoff and cryospheric elements including glacier, snow and frozen soil in a typical mountainous catchment. Overall, the manuscript is well structured and written and easy to follow. It is suitable for publication in HESS, especially for this special issue. However, I would like to point out two major concerns regarding the uncertainty and reliability of the results

Reply:

We thank Anonymous Referee #1 for positive remarks about our manuscript's structure and the suitability for publication in HESS. We have carefully considered all of your comments and provided responses or made necessary modifications to improve our manuscript. The following documents our detailed response to each of your comment.

1. The model validation is poorly conducted. Although the authors claimed that the parameters are adopted from a previous study in this catchment, some results related to model performance should be presented to show the confidence of model. If I understand correctly, the model in this study is the combination of the model in Gao et al. (2022) and the $\Delta$h-parameterization. Isn't there new parameter brought by this module compared to the previous version? Could the -Cryo model simulate something that cannot be simulated by -FS model, and if so, how does the model perform on simulating this additional objective? It is rather easy to simulate the relative change of glacier thickness, but simulating the absolute thickness of glacier is difficult, which significantly influences the conclusions such as the time glacier will disappear. So, again, please present some results to show reliability of glacier simulation. Even though all the simulations are the same with -FS model, some results need to be provided to show the confidence of model.

Reply:

We appreciate the reviewer's concern about the model validation and the need to demonstrate the reliability of the glacier simulation. We would like to clarify that the $\Delta$h-parameterization method used to calculate the glacier evolution is an empirical method based on the glacier melting calculated by the degree-day method. The same method for calculating the glacier melting is used in the FLEX-Cryo model and FLEX-FS model. So, no new parameters are introduced in the FLEX-Cryo model.

Regarding the glacier thickness (Fig. 7), we would like to state that Fig. 7 in the manuscript presents shows the change in absolute thickness, not the relative change of glacier thickness. The actual glacier thickness was calculated by subtracting the variation obtained from the Δh-parameterization method from the initial glacier thickness.

To address your concern about the model performance, we have included the results of runoff simulation in Figure 5. The model performance was evaluated using several metrics, including the KGE of 0.83, NSE of 0.73, R of 0.86, and RMSE is 0.77 mm/day for the period from 2011 to 2014. These results demonstrated that the FLEX-Cryo model can effectively reproduce observed hydrographs, indicating its ability to accurately predict future hydrological changes.

2. The uncertainty issue is addressed inadequately, although the authors mention it in the limitation section. I understand that this study aims to perform a systematic projection on the mountain cryosphere and hydrology, thus does not discuss much about the uncertainties of model parameter and GCM bias correction. However, I think the authors should at least report the uncertainties from different GCMs, given that eight GCMs are adopted for climate projection. The uncertainty range should be provided for the values in the main text (e.g., L304~314) and Figures (e.g., Figure 4).

Reply: Thanks for raising this important point about addressing uncertainties related to the use of GCMs from CMIP6. Indeed each GCM in CMIP6 demonstrates varying prediction capacities across different regions, resulting in a spectrum of uncertainties. To address your comment, we have made the following modifications in the revised manuscript.

Firstly, in Section 4.1.1, we have added an accuracy assessment of the eight GCMs used in this study by using Taylor diagram. Furthermore, we have added discussion on the reliability of the GCMs data after applying the bias correction. From the section 4.1.1 and Taylor diagrams, we can see that the bias correction improves the accuracy of each GCM in simulating historical temperature and particularly precipitation.

Secondly, we have updated Figure 6 to include the uncertainty ranges for projected temperature and precipitation based on the eight GCMs.

Thirdly, we have added the uncertainty range associated with predicted runoff in Figure 11, reflecting the propagation of GCM uncertainties through our hydrological model.

Other specific issues:

1. Please adjust the paragraph format to justified.

Reply: We have adjusted the paragraph format to justified in this revised manuscript.

2. Please provide a table to list the meaning of all the variables in Table 3 and Figure 3, to make them easier to find.

Reply: We have added Table 4 to list the meaning of all variables in figure 3 and Table3.

3. There are many GCMs in CMIP6. Why are these eight GCMs selected?

Reply: The eight Global Climate Models (GCMs) in CMIP6 were selected based on the previous validated studies. The eight GCMs have been well validated in the adjacent catchments by previous studies. Despite inherent regional simulation uncertainties associated with each GCM, we have applied widely used statistical downscaling, bias correction, and equal-weighted average methods to mitigate these uncertainties. By using well-validated GCMs and the widely used procedures for refining their outputs, we have enhanced the reliability of the FLEX-Cryo model results. We added the reasons for GCMs selection in the Section 2.2 (Line 179-211). We added the Taylor diagrams and discussion to assess the accuracy of each GCM in Section 4.1.1 in the revised manuscript.

I suggest the authors to reconstruct the Methodology section to make it more readable. It would be better to introduce the model first, and then introduce the spatial discretization of the catchment (the first paragraph of the current Methodology section), because the elevation band and HRU is the simulation unit of the model. Besides, more details of Δh-parameterization method need to be provided in the 3.1.1 section. The current description is too general, which is difficult to understand for a reader not familiar with this method.

Reply:Following your suggestion, we have reconstructed the Methodology section and added more details about the Δh-parameterization method in the revised manuscript. In Section 3, we have firstly introduced the FLEX-Cryo model including the glacier melting module, rainfall-runoff module, Δh-parameterization method and frozen soil module; then introduced the spatial discretization of the catchment; finally introduced the model evaluation metrics. In Section 3.3, we have added more details about the Δh-parameterization method including the selection and interpretation of empirical formulas to help the readers to understand how this method works.

4. Why a single value is provided for some parameters in Table 2, but a range is provided for others?

Reply:Sorry for this confusion. The parameter D is set as 0.2 from the isotope study (Ma et al., 2021). The precipitation increasing rate ($P_{calt}$) and (temperature lapse rate) $T_{calt}$ are set as 4.2 %/100 m and −0.68 ∘C/100 m based on the field measurements (Han et al., 2013). Respectively, The thermal conductivity (k), water content as a decimal fraction of the dry soil weight ( $\omega$ ) and bulk density of the soil ( $\rho$ ) are set as 2 W (m K)$^{-1}$, 0.12 and 1000 kg m$^{-3}$ , which have been verified by the simulated freezing depth using the Stefan equation (Gao et al., 2022). The other parameters with range were calibrated by Gao et al (2022) using the GLUE method. In this paper, the optional parameter set were selected based on Gao et al (2022) and the specific values of all parameters for the FLEX-Cryo model are added in Table 2.

Reference: Gao, H., Han, C., Chen, R., Feng, Z., Wang, K., Fenicia, F., and Savenije, H.: Frozen soil hydrological modeling for a mountainous catchment northeast of the Qinghai–Tibet Plateau, Hydrol. Earth Syst. Sci., 26, 4187–4208, https://doi.org/10.5194/hess-26-4187-2022, 2022.

**Responses to Anonymous Referee #2**

Climate change and cryosphere variation pose huge threats to local and downstream water resource security, and economic, social, and ecological sustainable development. However, there is still lack of integrated modeling tools to systematic project future changes of cryosphere and its impacts on hydrological regime in diverse climate change scenarios. Thus, this model projection study for the Hulu catchment in the Upper Heihe river has clear novelty. This topic fits well with the scope of this special issue on "Hydrological response to climatic and cryospheric changes in high-mountain regions". However, I agree with anonymous Reviewer 1 that the authors need to validate the proposed model and add uncertainty envelope in their results and figures, which are very important for the reliability of model and results. In addition, I have some minor concerns before the manuscript can be considered for publication.

Reply: We thank Anonymous Referee #2 for the providing positive remarks on the significance of our work and further constructive comments that help us improve our manuscript. We have carefully considered all comments and made necessary modifications in the revised manuscript. The following documents our detailed response to each of your comment.

1. In the abstract, two climate scenarios (SSP2-4.5 and SSP5-8.5) are not mentioned.

Reply: Now in the updated Abstract, we. have mentioned the two climate scenarios (SSP2-4.5 and SSP5-8.5) which is convenient for readers to understand the degradation of cryospheric elements and change of runoff under different scenarios.

2. Table 2. What are the optimized parameter values?

Reply:The optimized parameter values were calculated and obtained from Gao et al (2022) using the GLUE method. The updated Table 2 present the specific parameter values. We have added more necessary description about the optimized parameter values in the Section 3.1 in the revised manuscript.

3. Some important references are missing in the text. For example, there is lack of reference about the study site; no reference about CMIP6 dataset. Line 216~219, References are needed for the Δh parametrization. Some new model developments for small cold region catchments are not well cited, including but not limited to https://hess.copernicus.org/articles/27/4409/2023/; https://agupubs.onlinelibrary.wiley.com/doi/10.1029/2022WR033363;

Reply: Thanks for pointing out this important issue. To addess this issue, we have added a few more relevant references on the study area, CMIP6 dataset, the FLEX model construction and the △h-parametrization metho in the revised manuscript. The added new references are as follow:

Aubry-Wake, C. and Pomeroy, J. W.: Predicting Hydrological Change in an Alpine Glacierized Basin and Its Sensitivity to Landscape Evolution and Meteorological Forcings, WATER RESOURCES RESEARCH, 59, https://doi.org/10.1029/2022WR033363, 2023.

Chen, R., Duan, K., Shang, W., Shi, P., Meng, Y., and Zhang, Z.: Increase in seasonal precipitation over the Tibetan Plateau in the 21st century projected using CMIP6 models, Atmospheric Research, 277, 106306, https://doi.org/10.1016/j.atmosres.2022.106306, 2022.

Gao, H., Feng, Z., Zhang, T., Wang, Y., He, X., Li, H., Pan, X., Ren, Z., Chen, X., Zhang, W., and Duan, Z.: Assessing glacier retreat and its impact on water resources in a headwater of Yangtze River based on CMIP6 projections, Science of The Total Environment, 765, 142774, https://doi.org/10.1016/j.scitotenv.2020.142774, 2021.

Giovando, J. and Niemann, J. D.: Wildfire Impacts on Snowpack Phenology in a Changing Climate Within the Western U.S., Water Resources Research, 58, e2021WR031569, https://doi.org/10.1029/2021WR031569, 2022.

Hu, H., Ye, B., Zhou, Y., and Tian, F.: A land surface model incorporated with soil freeze/thaw and its application in GAME/Tibet, SCI CHINA SER D, 49, 1311–1322, https://doi.org/10.1007/s11430-006-2028-3, 2006.

Li, Z., Feng, Q., Chen, W., Wang, T., Cheng, Yan, G., Xiaoyan, G., Yanhui, P., Jianguo, L., Rui, G., and Bing, J.: Study on the contribution of cryosphere to runoff in the cold alpine basin: A case study of Hulugou River Basin in the Qilian Mountains, Global and Planetary Change, 122, 345–361, https://doi.org/10.1016/j.gloplacha.2014.10.001, 2014.

Liu, J. and Chen, R.: Discriminating types of precipitation in Qilian Mountains, Tibetan Plateau, Journal of Hydrology: Regional Studies, 5, 20–32, https://doi.org/10.1016/j.ejrh.2015.11.013, 2016.

Ma, J., Li, R., Huang, Z., Wu, T., Wu, X., Zhao, L., Liu, H., Hu, G., Xiao, Y., Du, Y., Yang, S., Liu, W., Jiao, Y., and Wang, S.: Evaluation and spatio-temporal analysis of

surface energy flux in permafrost regions over the Qinghai-Tibet Plateau and Arctic using CMIP6 models, International Journal of Digital Earth, 15, 1947–1965, https://doi.org/10.1080/17538947.2022.2142307, 2022.

Martin, L. C. P., Westermann, S., Magni, M., Brun, F., Fiddes, J., Lei, Y., Kraaijenbrink, P., Mathys, T., Langer, M., Allen, S., and Immerzeel, W. W.: Recent ground thermo-hydrological changes in a southern Tibetan endorheic catchment and implications for lake level changes, Hydrol. Earth Syst. Sci., 27, 4409–4436, https://doi.org/10.5194/hess-27-4409-2023, 2023.

Peng, Z., Tian, F., Wu, J., Huang, J., Hu, H., and Darnault, C. J. G.: A numerical model for water and heat transport in freezing soils with nonequilibrium ice-water interfaces, Water Resources Research, 52, 7366–7381, https://doi.org/10.1002/2016WR019116, 2016.

Xing, Z. P., Zhao, L., Fan, L., Hu, G.-J., Zou, D. F., Wang, C., Liu, S.-C., Du, E.-J., Xiao, Y., Li, R., Liu, G.-Y., Qiao, Y.-P., and Shi, J.-Z.: Changes in the ground surface temperature in permafrost regions along the Qinghai–Tibet engineering corridor from 1900 to 2014: A modified assessment of CMIP6, Advances in Climate Change Research, 14, 85–96, https://doi.org/10.1016/j.accre.2023.01.007, 2023.

Yin, G. A., Niu, F. J., Lin, Z.-J., Luo, J., and Liu, M.-H.: Data-driven spatiotemporal projections of shallow permafrost based on CMIP6 across the Qinghai–Tibet Plateau at 1 km2 scale, Advances in Climate Change Research, 12, 814–827, https://doi.org/10.1016/j.accre.2021.08.009, 2021.

Zhu, Y. Y. and Yang, S.: Evaluation of CMIP6 for historical temperature and precipitation over the Tibetan Plateau and its comparison with CMIP5, Advances in Climate Change Research, 11, 239–251, https://doi.org/10.1016/j.accre.2020.08.001, 2020.

4. It is better to add a landscape classification map in Figure 1.

Reply: We have added new maps for the spatial distribution of four landscapes in Figure 1 (glacier, alpine desert, vegetation hillslope, and riparian zone).

5. Line 201~204, the equal weighted average method could be more clearly demonstrated by equations. Please give specific functions.

Reply: We have added this equation in the 2.2 section.

$$P_{ave} = \frac{1}{N_{GCM}} (\sum_{j=1}^{N_{GCM}} (\frac{1}{N_{bias}} (\sum_{i=1}^{N_{bias}} (P_i))))$$

where, the $P_{ave}$ is the average value of the multi-model and multi-method, $P_i$ is the projected climate data of an GCM, $N_{bias}$ is the number of correction methods ($N_{bias}$ is 3 in this research) and NGCM is the number of GCM ($N_{GCM}$ is 8 in this study).

6. Some figures are not mentioned in the main text, such as Figure 2, 3.

Reply:Figure 2 shows the landscape classification at different elevation bands and Figure 3 shows the structure of the FLEX-Cryo model. In this study, glacier, alpine desert, hillslope vegetation and riparian zone landscapes were identified. We have added the description of Figure 2 and Figure 3 in Section 3.

7. Line 235. How did you calculate the snow cover days and snow water equivalent?

Reply: In the FLEX-Cryo model, the $S_w$ is the snow pack reservoir. The maximum value of the $S_w$ in a year represents the snow water equivalent and the number of the non-zero value is the snow cover days. We have added more details regarding these in Section 3.1.1 in the revised manuscript.

8. Line 375. How did you calculate the lower limit of permafrost?

Reply: F is the surface freeze/thaw index, which represents the cumulative value of the temperature below (above) 0℃ (equation (18)). At the lower limit of permafrost, the freeze index is equal to the thaw index. On the permafrost area, the freeze index is greater than thaw index, and on the seasonal frozen soil area is opposite. We have added the description in Section 3.2 in the revised manuscript.

9. The conclusion can be shortened.

Reply: Thanks for your suggestion. We shortened our conclusion section (in section 6). And the shorten conclusion is as follow:

In this study, we employed the FLEX-Cryo model and data from eight Global Climate Models (GCMs) under the SSP2-4.5 and SSP5-8.5 scenarios to predict the potential impacts of climate change on the mountain cryosphere and hydrology. Results from the projected change of mountain cryosphere elements, glacier, snow and frozen soil are expected to undergo degradation. The glacier will completely melt by the middle of the 21st century. Snow cover day will decrease by 45 and 76 days, and snow water equivalent will decrease by 0.24mm/yr and 0.35mm/yr. The thaw

onset is expected to advance 19 days and 32 days. The active layer thickness will increase by 8.24cm/10yr.

The degradation of the mountain cryosphere has significant implications for water resources. The tipping point of glacier runoff is projected to occur in the 2020s. Once the glaciers have completely melted, the runoff is projected to decrease by approximately 16% and 18% under the SSP2-4.5 and SSP5-8.5 scenarios, respectively. Importantly, the duration of low runoff will shorten, baseflow will increase and the discontinue recession of baseflow will gradually transform to a more linear pattern.

This study provides insights into the potential impacts of climate change on the mountain cryosphere and hydrology. The projected changes in glacier retreat, snow cover, and frozen soil dynamics highlight the urgent need for proactive water resource management strategies in the face of a changing climate. Further modelling research and monitoring efforts are necessary to refine these projections and guide effective adaptation measures to sustainably manage water resources in mountainous regions.

10. Some figures have small font size, e.g. Figure 1, 7, 8, and 11.

Reply: The font size has been enlarged in Figure 1, 9 10 and 12 in the revised manuscript.

**Responses to Anonymous Referee #3**

This work presents projection of future changes in glacier and runoff in the Upper-Heihe River. I have some major concerns about the methods, as follows, which the authors may consider addressing in the revision:

Reply: We thank Anonymous Referee #3 for providing us with constructive comments that help us improve our manuscript. We have carefully considered all comments and made necessary modifications in the revised manuscript. The following documents our detailed response to each of your comment.

1. Model Evaluation: While the authors claim the model has been evaluated in previous works, the newly integrated (or refined) glacier module could impact simulated runoff. I suggest the authors conduct a comprehensive evaluation of the model in simulating discharge, glacier, snow and soil water.

Reply: Thanks for pointing out this important issue. To address this issue, we have added more necessary model evaluation and results in the revised manuscript.

We have emphasized that while the framework of the FLEX-Cryo model remains largely unchanged compared to the FLEX-FS model; the runoff is still composed of the glacier melting, runoff in alpine desert, in vegetation hillslope and in riparian zone. The only update in the FLEX-Cryo model is that we refined the glacier evolution process. However, the glacier melting ($M_g$) calculation, which contributes to runoff, is still based on the degree-day method. $F_{dd}$ and $C_g$ are the specific parameter. The generated process $M_g$ has nothing changed (Eq. 1). $M_g$ is then, together with $P_l$ routed through a linear reservoir $S_g$ (Eq. 2), controlled by a recession parameter $K_{f,g}$, to compute the runoff generated ($Q_g$) from glacier areas (Eq. 3). So, the introduced Δh-method does not directly affect the glacier runoff yield process.

$$M_g = \begin{cases} F_{dd} \cdot T \cdot C_g & S_w \; and \; T > 0 \\ 0 & S_w \; and \; T > 0 \end{cases} \quad (1)$$

$$\frac{dS_g}{dt} = P_l + M_g - Q_g \quad (2)$$

$$Q_g = \frac{S_g}{K_{f,g}} \quad (3)$$

where, $F_{dd}$ (mm°C$^{-1}$d$^{-1}$) is snow degree day factor, $C_g$ (-)is the glacier degree factor multiplier, T (℃) is air temperature above the threshold temperature, $M_g$ (mm/day) is

glacier runoff, $S_g$ (mm) is the glacier reservoir, $P_l$ (mm/day) is rain, $Q_g$ (mm/day) is discharge from the glacier, $K_{f,g}$ (day) is the recession parameter for glacier.

In Section 4.1.2, we have added the performance metrics of the FLEX-Cryo model in simulating runoff. The results show a good performance with a KGE of 0.83, NSE of 0.73, R of 0.86, and RMSE of 0.77 mm/day. In this study, we selected the optimal parameters group from the parameter set in our previous study Gao et al. (2022), which may result in the small changes of model evaluation metrics. We have clarified that no new parameters were introduced in the FLEX-Cryo model compared to the FLEX-FS model, and the selected parameters are still based on the research by Gao et al. (2022). This, combined with the unchanged model framework, suggests that the newly refined glacier module does not have a significant impact on simulated runoff.

2. Parameterization of the Δ-h Module: The Δ-h module, originally developed and applied in Alpine glaciers in Switzerland, adopted empirical parameter values from long-term observations of glacier area over Switzerland. Its practical application in China has not been well evaluated. The authors should provide more details on the verification of this module in their study area, as well as on the determination of parameter values.

Reply: To address your comment, we have made the following modifications in the revised manuscript. We have clarified that the Δh-parameterization method has been widely applied not only in Alpine glaciers in Switzerland but also in various regions of China, such as Urumqi Glacier No. 1, Dongkemadi Glacier catchment, and the headwater of the Yangtze River Basin (references). These studies have demonstrated its good simulation ability and practical application in China.

We have emphasized that the Δh-module is used to calculate glacier evolution rather than glacier melting. The selection of the empirical equation for the Δh-parameterization is based on glacier size. In our study, both Glacier 1 and Glacier 2 are small glaciers (area < 5 km2). Therefore, we have used the following empirical equation (Eq. 4):

$$\Delta h = (h_r - 0.30)^2 + 0.60(h_r - 0.30) + 0.09 \quad (4)$$

Where, Δh is normalized surface elevation change and $h_r$ is the normalized elevation range.

As also mentioned in our response to your previous comment, In Section 4.1.2, we have added the performance metrics of the FLEX-Cryo model simulation. Throughout the

entire assessment period, the model shows good simulation performance with a KGE of 0.83, NSE of 0.73, R of 0.86, and RMSE of 0.77 mm/day. These results further support the applicability of our FLEX-Cryo model in our study area.

3. Coarse Spatial Resolution of CMIP Model Products: This study was conducted in a small basin with 23km$^2$, but the CMIP products were only downscaled to a resolution of 0.5 deg, much larger than the basin size. To reduce uncertainty, downscaled inputs at higher resolutions would be beneficial.

Reply: We acknowledge that the spatial resolution of the CMIP Model products (0.5 deg) is larger than the size of our study basin (23 km$^2$). To mitigate the uncertainty associated with this scale mismatch, we have applied widely used statistical downscaling, bias calibration, and equal-weighted average methods to the CMIP6 GCM outputs. These processing steps have indeed increased the accuracy of the precipitation and temperature estimates from each GCM. We have added more details about our processing and analysis of the uncertainty in Section 2.2, and Taylor diagrams have been added in Section 4.1.1 to discuss the improved accuracy of GCM outputs after applying the bias correction.

4. Uncertainty in the Analysis: The results inevitably imply significant uncertainty from model inputs, parameters, and assumptions. A full assessment of modeling uncertainty is highly recommended.

Reply: We agree with you. To address this comment, we have added the uncertainty ranges of temperature and precipitation in Figure 6, and the uncertainty ranges of runoff in Section 4.3.4. The uncertainty in Figure 11 suggests that glacier melting may play an important role in supplying runoff, especially before the tipping point of glacier melting. We have also added more discussion on uncertainty and limitations in Section 5.3 in the revised manuscript.

5. Sharp Decreases in Glacier Thickness after 2040: In figure 5, both glaciers exhibit a sharp decrease in thickness but only small changes in volume after 2040. More explanation is needed, as such sharp changes in climate are not observed.

Reply: Thank you for this excellent question. To answer your question here, Figure 7 (a) shows the change of glacier thickness at the highest elevation band and Figure 7 (b) shows the change of overall glacier volume for Glacier 1 or Glacier 2. Therefore, due to the different focus of these two figures, the timing of sharp changes may not coincide for glacier thickness and glacier volume. In this study, we divided the elevation into 37 bands and distributed the landscape to each elevation band. Before 2040, based on the $\Delta$ h-parameterization method (Equation 18), if glaciers span more than one elevation

band, the $h_r$ (normalized elevation range parameters) would be equal to 1 at the lowest elevation band where the glacier thickness change faster. After 2040, the glacier only exists in the highest elevation band, where the $h_r$ increase to 1 and glacier thickness changes fast. However, at this point, most of the glacier will have already melted, leading to a small change in glacier volume. We have added this discussion in Section 5.1 in the revised manuscript to clarify the observed patterns.

6. Change 'day' to DOY in the y-axis of Figures 6a-b.

Reply: We have changed this in Figure 8 in the revised manuscript.

7. Conclusion: the conclusion is wordy and could be more straightforward and information

Reply: We have shorten the Conclusions Section. The updated Conclusions read as follows:

In this study, we employed the FLEX-Cryo model and data from eight Global Climate Models (GCMs) under the SSP2-4.5 and SSP5-8.5 scenarios to predict the potential impacts of climate change on the mountain cryosphere and hydrology. Results from the projected change of mountain cryosphere elements, glacier, snow and frozen soil are expected to undergo degradation. The glacier will completely melt by the middle of the 21st century. Snow cover day will decrease by 45 and 76 days, and snow water equivalent will decrease by 0.24mm/yr and 0.35mm/yr. The thaw onset is expected to advance 19 days and 32 days. The active layer thickness will increase by 8.24cm/10yr.

The degradation of the mountain cryosphere has significant implications for water resources. The tipping point of glacier runoff is projected to occur in the 2020s. Once the glaciers have completely melted, the runoff is projected to decrease by approximately 16% and 18% under the SSP2-4.5 and SSP5-8.5 scenarios, respectively. Importantly, the duration of low runoff will shorten, baseflow will increase and the discontinue recession of baseflow will gradually transform to a more linear pattern.

This study provides insights into the potential impacts of climate change on the mountain cryosphere and hydrology. The projected changes in glacier retreat, snow cover, and frozen soil dynamics highlight the urgent need for proactive water resource management strategies in the face of a changing climate. Further modelling research and monitoring efforts are necessary to refine these projections and guide effective adaptation measures to sustainably manage water resources in mountainous regions.

---

## Author Response (AR2)

Many thanks to the authors for revising the manuscript. Most of my concerns have been addressed in the revised version. However, although the authors have added validation of model performance, only the performance on streamflow simulation is presented. As known to us, the simulation of cryospheric factors could have large uncertainties even if the streamflow is simulated well. Consequently, I would like to recommend another round of revision, to add the simulations of glacier, snow and frozen soil during historical period, and the validation of these simulations by observation data.

Reply:

We appreciate the reviewer's concern about the model validation and the need to demonstrate the reliability of the cryospheric factors. We have added the validation of the glacier mass balance and freeze-thaw cycle based on observation data (Section 4.1.2). The updated results as follows:

We assessed the performance of the FLEX-Cryo model for glacier mass balance change, freeze/thaw depth and runoff simulation based on historical observations. The model demonstrated strong capabilities across all evaluated aspects. For the glacier mass balance change, the model showed good accuracy throughout the entire assessment period. Monthly simulations yielded a KGE value of 0.45, NSE of 0.83, the correlation coefficient R of 0.95 and RMSE of 130.13 mm/month (Figure 1a and Table 1). Regarding the free/thaw dynamics, the model accurately captured both timing and duration. The simulated freeze onset consistently aligned with observations, typically occurring in late October and early November. Moreover, the simulated freeze-thaw cycle duration closely matched observations, with both spanning approximately 217 days and varying by no more than 15 days. Notably, the model exhibited exceptional accuracy in predicting maximum freezing depth, with a mere 2 mm error recorded in April 2013 (Figure 1b and Table 1).

[Figure]

Figure.1 (a) Comparison of the modelled and observed glacier mass balance (GMB) of Glacier 1 from Jan. 2011 to Dec. 2014. (b) comparison of the simulated freeze/thaw depth by Stefan equation and observation.

Table 1. The results of evaluation metrics

|  | KGE | NSE | R | RMSE |
| --- | --- | --- | --- | --- |
| Glacier mass balance | 0.45 | 0.83 | 0.95 | 130.13 mm/month |
| Runoff depth | 0.83 | 0.73 | 0.74 | 0.77 mm/day |

**Anonymous Referee #2**

The authors have well addressed and modified the part related to model evaluation and uncertainty analysis, which greatly improves the quality of the manustript in strucutre and reliability of manuscript.

Reply: Thank you for your constructive comments that help us improve the quality of our manuscript.